# ZERO-SHOT GOAL DIALOGUE VIA REINFORCEMENT LEARNING ON IMAGINED CONVERSATIONS

## ABSTRACT

Large language models (LLMs) have emerged as powerful and general solutions to many natural language tasks. However, many of the most important applications of language generation are interactive, where an agent has to talk to a person to reach a desired outcome. For example, a teacher might try to understand their student's current comprehension level to tailor their instruction accordingly, and a travel agent might ask questions of their customer to understand their preferences in order to recommend activities they might enjoy. LLMs trained with supervised fine-tuning or "single-step" RL, as with standard RLHF, might struggle which tasks that require such goal-directed behavior, since they are not trained to optimize for overall conversational outcomes after multiple turns of interaction. In this work, we explore a new method for adapting LLMs with RL for such *goal-directed dialogue*. Our key insight is that, though LLMs might not effectively solve goal-directed dialogue tasks out of the box, they can provide useful data for solving such tasks by simulating human-like behaviors. Given a textual description of a goal-directed dialogue task, we leverage LLMs to synthesize hypothetical in-domain human-human interactions. Our algorithm then utilizes this dataset with *offline reinforcement learning* to train an interactive conversational agent that can optimize multi-step objectives. Empirically, we show that our proposed approach achieves state-of-the-art performance in various goal-directed dialogue tasks that include teaching and preference elicitation.

## 1 INTRODUCTION

Large language models (LLMs) have become very effective at performing a variety of real-world natural language tasks, including open-ended question-answering (Pyatkin et al., 2022), summarization (Paulus et al., 2017; Wu & Hu, 2018; Böhm et al., 2019), code generation (Chen et al., 2021b; Rozière et al., 2023; Zhong & Wang, 2023), and general problem-solving (Wei et al., 2023). While LLMs shine at producing compelling and accurate responses to individual queries, their ability to engage in *goal-directed* conversation remains limited. They can

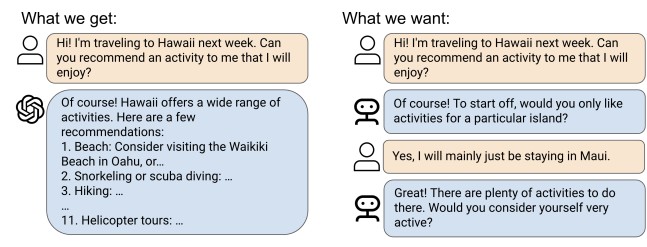

Figure 1: Illustrative example of how existing LLMs behave when prompted to act as travel agents vs. how effective travel agents should behave (what our method ultimately achieves).

*emulate* the flow of a conversation, but they generally do not aim to accomplish a goal through conversing. For example, and as shown in Figure 1, we can prompt an LLM to act as a travel agent, and it will produce realistic responses that a human may rate as helpful. But it will not intentionally try to maximize the chance of planning a desirable itinerary for the human. In practice, this manifests as a lack of clarifying questions, lack of goal-directed conversational flow, and generally verbose and non-personalized responses.

The difference between an agent that simply mimics the flow of a conversation and one that pursues conversational goals becomes more apparent when we consider how *uncertainty* influences the

conversation. Whether you as the end user are asking the agent to instruct you about a new AI concept, or to plan an itinerary for an upcoming vacation, you have priviledged information which the agent does not know, but which is crucial for the agent to do the task well; e.g., your current background of AI knowledge matters when learning a new concept, and your travel preferences matter when you plan a vacation. As illustrated in Figure 1, a *goal-directed* agent would gather the information it needs to succeed, perhaps by asking clarification questions (e.g., are you an active person?) and proposing partial solutions to get feedback (e.g., how does going to the beach sound?). However, today's LLMs largely fail at this, and are more likely to attempt a helpful but poorly informed guess right away than to ask appropriate questions. And as we will show in the experiments, even when carefully prompted to gather information, they comply but generate verbose and overwhelming questions that are not good at getting the right information.

In principle, reinforcement learning (RL) can offer a very powerful tool for bridging this gap: LLMs trained with RL to achieve conversational goals (such as maximizing the probability that the user will accept the planned itinerary) could take goal-directed steps, ask clarifying questions, elicit preferences, be very clear and concise in its responses, and maybe even build a rapport with the user. But RL requires data, either in the form of online interactions with a human simulator, or offline human-human interactions. Online data can be computationally difficult obtain, and offline data must be carefully curated to optimize desirable properties such as coverage and diversity (Fu et al., 2020b; Gulcehre et al., 2020; Kumar et al., 2022).

Our key idea is that we can enable *zero-shot goal-directed dialogue agents* by tapping into what LLMs are great at — emulating diverse realistic conversations; and tapping into what RL is great at — optimizing multi-step objectives. We propose to use LLMs to "imagine" a range of possible task-specific dialogues that are often realistic, but where the LLM does not optimally solve the task. In effect, the LLM can imagine what a human *could* do, but not to what an optimal agent *should* do. Conversations are then generated based on sampled hidden states. We train an agent to engage in goal-directed conversation by training offline RL on the resulting dataset.

Our main contribution is a zero-shot RL algorithm that effectively optimizes for goal-directed dialogue tasks. Rather than directly using pretrained LLMs as optimal agents, our method aims to leverage their strength in emulating diverse, human-like, but suboptimal conversations to generate data, which can then be provided to an RL algorithm to actually discover more optimal behaviors. We propose a novel system called the *imagination engine* (IE) that generates a dataset of diverse, task-relevant, and instructive dialogues to be used to train downstream agents. We evaluate our approach on tasks involving teaching of a new concept, persuasion, and preference elicitation. Our experimental results include a user study that compares agents trained with our method to prompted state-of-the-art LLMs, showing that our method can attain significantly better results in interactive conversations even when using models that are orders of magnitude smaller than the prompt-based baseline.

## 2   RELATED WORK

**Language models.** Language models, particularly LLMs, have shown impressive capabilities in text generation (Ghazvininejad et al., 2017; Li et al., 2017; Holtzman et al., 2018; Radford et al., 2019; Yang & Klein, 2021), translation (Gu et al., 2017), question answering (Pyatkin et al., 2022), summarization (Paulus et al., 2017; Wu & Hu, 2018; Böhm et al., 2019), and code generation (Chen et al., 2021b; Zhong & Wang, 2023). However, success at most of these tasks is largely enabled by supervised learning, and does not require reasoning through multiple steps of interaction to optimize a long-term objective. LLMs have been fine-tuned via supervised learning to engage in dialogue with human users to some success (He et al., 2018; Shuster et al., 2022b;a), but primarily to only produce realistic responses

**RL for language models.** Many existing LLMs leverage reinforcement learning (RL) fine-tuning, where a reward model is learned from feedback directly from human experts (Ziegler et al., 2020; Stiennon et al., 2020; Wu et al., 2021; Nakano et al., 2022; Bai et al., 2022a; Christiano et al., 2023) or secondhand from a handcrafted AI system (Bai et al., 2022b), and is then used to fine-tune the LLM via an RL objective. While finetuning is primarily done via online RL, recent approaches proposed tuning LLMs from offline data  (Rafailov et al., 2023; Gulcehre et al., 2023). By doing so, LLMs are able to faithfully follow human instructions, or *prompts*, and can therefore act as general problem solvers by prompt engineering (Ouyang et al., 2022). While effective, one stark downside of RL fine-tuning approaches is that they only consider bandit objectives. Specifically, in RL fine-tuning,

LLMs are trained to maximize the learned reward model within a single-step response, and not over the course of a multi-step dialogue. As a result, if the best response to a query is unknown due to latent information, such as intentions or preferences, by the user, traditional LLMs will only provide the best possible "guess" response in one step, and not attempt to gather additional information in order to respond more optimally. Notably, Glaese et al. (2022) propose learning an information-seeking agent, but again consider a single-step objective based on maximizing helpfulness, and do not consider nor evaluate on tasks where gathering information is used to accomplish a long-term goal; the approach also relies on human raters being able to identify useful information-seeking actions.

**Goal-directed dialogue.** There has been numerous prior works on learning models to accomplish tasks via conversations beyond maximizing informativeness or humanness. Goal-directed dialogue, or alternatively task-oriented dialogue, can be formulated as an MDP from which agents can be trained using RL. Online RL methods to optimize dialogue agents typically require a simulator of human behavior, that is usually either handcrafted, or learned as a fixed model (Carta et al., 2023; He et al., 2018; Gašić et al., 2011). Moreover, they involve continual collection of new samples, which incurs a large computational cost in tasks where humans exhibit complex and nuanced behaviors, and are often prone to reward "hacking" (Skalse et al., 2022). Alternatively, offline RL approaches have also been considered that only require a static dataset of dialogues (Jaques et al., 2019; Jang et al., 2022; Verma et al., 2022; Snell et al., 2023). Notably, Verma et al. (2022) propose an offline RL algorithm to solve a goal-directed dialogue based on negotiations using a dataset of conversations between human speakers. However, in order for offline RL to improve over supervised learning, the dataset must be carefully curated to optimize desirable properties such as coverage and diversity (Fu et al., 2020b; Gulcehre et al., 2020; Kumar et al., 2022), which may limit its practicality. Orthogonally, recent approaches to learning dialogue agents often do not use RL, but rely on natural language instructions, or prompts, to guide LLMs towards accomplishing various tasks. These prompts are often manually engineered (Brown et al., 2020; Reynolds & McDonell, 2021; Wei et al., 2022), or more recently, optimized via supervised learning (Deng et al., 2022; Li et al., 2023). However, we believe prompt-engineering is insufficient to solving general goal-directed dialogue tasks, especially when desired behaviors are difficult to describe using only task-specific instructions and demonstrations. Furthermore, prompt-learning primarily has only considered single-step objectives and often requires access to human annotations of hints.

**Dialogue benchmarks.** Dialogue benchmark datasets have been created that aim to evaluate the capabilities of agents at accomplishing various tasks such as question-answering (Budzianowski et al., 2020), customer service (Chen et al., 2021a), and negotiation (He et al., 2018). However, many such datasets are for tasks that do not necessitate personalizing the agent's responses to each human. In this paper, we consider goal-directed dialogue tasks where humans behave differently due to latent factors, and agents must gather information and personalize to each human. Because of this added complexity, curating a human-human dataset with diverse human behaviors can be prohibitively difficult.

**Knowledge distillation.** Our proposed imagination engine can be considered an instance of knowledge distillation (Hinton et al., 2015), where knowledge from a large model (in our case, LLMs) is used to train a smaller model. Recently, this has become popular with LLMs acting as the teacher model, and synthetically generating new training examples for the smaller model (Taori et al., 2023; Chiang et al., 2023; Kim & Rush, 2016). While our approach is similar in principle, all prior approaches consider only downstream supervised learning objectives. To our knowledge, we are the first to do synthetic dialogue generation for RL.

## 3 PRELIMINARIES

**Markov decision processes.** To formulate dialogue as a decision making problem, we use the formalism of the Markov decision process (MDP), given by a tuple $M = (\mathcal{S}, \mathcal{A}, P, r, \rho, \gamma)$, where $\mathcal{S}$ is the state space, $\mathcal{A}$ is the action space, $P$ is the transition function, $r$ is the reward function, $\rho$ is the initial state distribution, and $\gamma$ is the discount factor. When action $a \in \mathcal{A}$ is executed at state $s \in \mathcal{S}$, the next state is sampled $s' \sim P(\cdot|s, a)$, and the agent receives reward $r$ with mean $r(s, a)$.

**Goal-directed dialogues as MDPs.** Goal-directed dialogue can be viewed as an MDP, where states are sequences of tokens from a finite vocabulary $\mathcal{V}$ (Ramamurthy et al., 2023). All tokens that the agent initially observes are used as our initial state, $s_0 = (x_0, \ldots, x_m)$, where $x_i \in \mathcal{V}, \forall i \in [m]$. At timestep $t$, an action $a_t \in \mathcal{V}$ is some token in the vocabulary. As long as $a_t$ is not a special

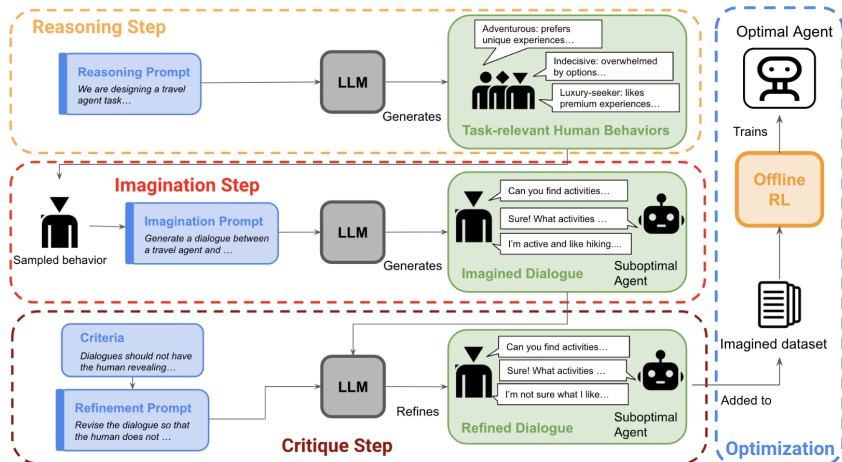

Figure 2: Diagram illustrating our proposed approach, where an imagined dataset of dialogues between humans and a potentially suboptimal agent is synthesized by our imagination engine, then used to train a downstream RL agent. Blue boxes indicate handcrafted quantities.

end-of-sequence <EOS> token, the transition function deterministically appends $a_t$ to state $s_t$ to form $s_{t+1}$. Otherwise, the agent observes (potentially stochastic) responses from their conversational partner $o_t = (y_0, \ldots, y_n)$, which also consist of tokens in the vocabulary; then, the transition function appends both $a_t$ and responses $o_t$ to state $s_t$. This continues until the last timestep $T$ where we obtain a state $s_T$ and the agent receives a deterministic reward $r(s_T)$.

In many real-world tasks that require dialogue with a human, humans exhibit a range of different behaviors. For example, in a travel agent task, humans will respond differently to the agent according to their own activity interests, budget, and other personal factors. Such factors are often latent, but affect how an optimal agent should respond. Rather than conventional MDPs, these tasks can instead be formulated as hidden parameter MDPs (Doshi-Velez & Konidaris, 2013), given by a tuple $M = (\mathcal{S}, \mathcal{A}, \mathcal{Z}, P, r, \rho, \gamma)$, where $\mathcal{Z}$ also parameterizes the transition and reward functions. In practice, solutions to hidden parameter MDPs do not need to model $\mathcal{Z}$ explicitly, and instead use a sequence model (i.e., a standard language model) to handle implicitly infer it from the history of observations. Nevertheless, we view $\mathcal{Z}$ as helpful formalism for understanding why information-gathering is important in effective dialogue agents.

**Reinforcement learning.** The goal of reinforcement learning (RL) is to learn a policy $\pi$ that maximizes the expected discounted return $\sum_{t=0}^{\infty} \gamma^t r_t$ in an MDP. The Q-function $Q^\pi(s, a)$ for a policy $\pi$ represents the discounted long-term reward attained by executing $a$ given state $s$ and then following policy $\pi$ thereafter. $Q^\pi$ satisfies the Bellman recurrence:

$$Q^\pi(s, a) = r(s, a) + \gamma \mathbb{E}_{s' \sim P(\cdot|s,a), a' \sim \pi(\cdot|s')} \left[ Q(s', a') \right]$$

The value function $V^\pi$ is the expectation of the Q-function $V^\pi(s) = \mathbb{E}_{a \sim \pi(\cdot|s)} \left[ Q^\pi(s, a) \right]$. The expected discounted return can be expressed as $J(\pi) = \mathbb{E}_{s_0 \sim \rho} \left[ V^\pi(s_0) \right]$. In offline RL, we are provided with a dataset $\mathcal{D} = \{(s_i, a_i, s_i', r_i)\}_{i \in [N]}$ of size $|\mathcal{D}| = N$, generated by an unknown behavior policy $\pi_\beta$ (which might correspond to a mixture of multiple policies). The offline RL setup is particularly useful when online interaction with the real world is costly or unavailable.

## 4 REINFORCEMENT LEARNING ON IMAGINED CONVERSATIONS

In this paper, we present an approach for *zero-shot* training of agents in a goal-directed dialogue task. Rather than traditional offline RL approaches that require a curated dataset $\mathcal{D}$ of human-human data, the only input required by our system is a task description $D$. The primary novelty of our proposed approach is an *imagination engine* (IE) that enables the generation of a diverse dataset $\widehat{\mathcal{D}}$ of task-relevant dialogues for any task description $D$. Then, once the dialogue dataset $\widehat{\mathcal{D}}$ is generated, we optimize for an agent $\widehat{\pi}$ via offline RL on the imagined dialogues.

## 4.1 IMAGINATION ENGINE: GENERATE DIVERSE DIALOGUES

We assume access to a LLM $P_{\mathsf{LLM}}(\cdot \mid p)$ that can be used to generate a response for any prompt $p$. The IE consists of three steps, which are outlined in Figure 1 and we describe below. We also provide explicit examples of the process (including prompts used) for two different tasks in Appendix A. Though our approach does rely on some prompting-engineering, we show that the prompts used are quite natural and not overly-engineered. Furthermore, in Appendix A, we show some qualitative evidence that different paraphrases of prompts do not noticeably affect the generation quality.

**Reasoning step: synthesize in-domain personas.** Recall that goal-directed dialogues can be formulated as hidden-parameter MDPs with hidden space $\mathcal{Z}$, where each human has a different $z \in \mathcal{Z}$ that affects how they behave, and how an agent should optimally respond. Without access to prior data, we would naïvely rely on having task-specific knowledge of $\mathcal{Z}$. However, our insight is that LLMs contain a much wider domain of knowledge than any individual human, and therefore can provide task-specific knowledge when humans cannot. Therefore, we propose querying $P_{\mathsf{LLM}}(\cdot \mid f_r(D))$, where $f_r(D)$ is a *reasoning prompt* using task description $D$; the prompt asks the LLM to output textual descriptions of personas $\phi(z)$ for $z \in \mathcal{Z}$. These descriptions $\phi(z)$ can be used to generate human responses under different behaviors (Serapio-García et al., 2023; Park et al., 2023).

**Imagination step: generate synthetic dialogues.** The goal in this step is to imagine dialogues $\tau$ between a (potentially suboptimal) agent and a human. Formally, this involves generating synthetic rollouts in the underlying hidden parameter MDP. Note that in real world, samples from both the transition function $P$ and behavior policy $\pi_\beta$ of the MDP are simply human responses. Therefore, synthesizing trajectories reduces to simulating human-human dialogue.

In order to accomplish this, we leverage LLMs to generate synthetic dialogues between an agent and human, where we condition generation on how the human behaves, and the reward that the agent achieves. This is done as follows. First, we sample $\phi(z)$ for some persona $z \in \mathcal{Z}$ that we obtained in the reasoning step, and also sample $r \in \{0, 1\}$ indicating whether the agent fails or succeeds in the generated dialogue. The assumption of binary rewards is only to be task-agnostic, and more expressive rewards can be considered if they exist for a particular task. A conditional dialogue can be sampled $\tau \sim P_{\mathsf{LLM}}(\cdot \mid f_i(D, \phi(z), r))$ where $f_i(D, \phi(z), r)$ is an *imagination prompt* that asks the LLM to output a dialogue between two humans that is task-relevant, and where the human behaves according to $z$ and the agent ultimately achieves reward $r$.

**Critique step: refine dialogues.** Though the synthetic dialogues from the imagination step are mostly plausible interactions, the humans in the dialogue sometimes produce potentially unrealistic responses. For example, humans in the dialogues often reveal their underlying personas, without the agent asking any questions, or even building general rapport with them. Since inferring the human's persona is an important skill we want downstream learning agents to acquire, we want information-gathering strategies to be reflected in the imagined conversations, even if they are not deployed optimally or strategically (as would be required for the optimal agent).

To remedy this, we propose revising the imagined dialogues based on a set of criteria $c$ on what constitutes pedagogical conversations for our downstream learning. Our criteria $c$ are task-specific but generally include the following principles: (1) the "human" should not reveal their latent behavior immediately, but only make it apparent gradually through back-and-forth conversation with the agent; (2) the human's sentiment at the end of the dialogue should accurately reflect the reward that the agent achieves. Our criteria can be used analogously to a constitution to improve the quality of generated responses (Bai et al., 2022b). Formally, we sample a revised dialogue $\tau' \sim P_{\mathsf{LLM}}(\cdot \mid f_c(D, \tau, c))$ from the same LLM where *critique prompt* $f_c(D, \tau, c)$ encapsulates the original dialogue and criteria $c$. Iterating the imagination and critique steps allows us to curate a dialogue dataset $\widehat{\mathcal{D}}$.

## 4.2 RL OPTIMIZATION ON THE IMAGINED DATASET

While the imagination engine can produce plausible dialogues, this does not by itself produce effective agents – that is, we use LLMs to synthesize plausible scenarios, including strategies that an agent may take, but not necessarily what an optimal agent should do. In order to determine the optimal strategy that an agent should actually take to achieve a desired outcome, we require multi-step RL to optimize an agent to maximize probability of succeeding at the desired task. Therefore, the main question we aim to answer in this section is the following: *How do we use a static dataset of synthetic*

*dialogues to train an RL agent?* Our solution involves running offline value-based RL to learn a policy purely from the synthetic dataset.

Before running offline RL, we need to postprocess the dataset of synthetic dialogues into RL training examples. Recall that we constructed a dataset $\widehat{\mathcal{D}} = \{(\tau_i, r_i)\}_{i \in [N]}$ of $N$ imagined dialogues, where each dialogue $\tau_i$ a sequence of tokens in vocabulary $\mathcal{V}$ that constitute utterances between a simulated agent and simulated human. For each dialogue $\tau_i$, we isolate all tokens $a$ by the agent, then generate $(s, a, s', r)$ where state $s$ consist of all tokens before $a$, next state $s'$ consist of all tokens before the next token $a'$ by the agent, and $r = r_i$ only if $s' = \tau_i$ is the full dialogue. Using this procedure, we construct a dataset $\widehat{\mathcal{D}}' = \{(s_i, a_i, s'_i, r_i)\}_{i \in [N']}$.

Then, we run value-based RL to learn a policy $\widehat{\pi}$. Specifically, we learn $\widehat{Q}$ and $\widehat{V}$ functions that estimate the optimal $Q$-function and value function, respectively, and then use these functions to extract a policy $\widehat{\pi}$. The functions can be learned using Bellman recurrence:

$$\widehat{Q} = \arg\min_Q \mathbb{E}_{(s,a,s',r) \sim \widehat{\mathcal{D}}'} \left[ \left( r + \gamma \widehat{V}(s') - Q(s,a) \right)^2 \right], \quad \widehat{V} = \arg\min_V \mathbb{E}_{s \sim \widehat{\mathcal{D}}'} \left[ \left( \max_{a'} \widehat{Q}(s,a') - V(s) \right)^2 \right].$$

When $\widehat{\pi}$ is a language model, we use these functions in combination with the base language model $\widehat{\pi}_{\mathsf{LM}}$ to extract the policy (Snell et al., 2022), via $\widehat{\pi}(a|s) \propto \pi_\beta(a|s) e^{\beta(\widehat{Q}(s,a) - \widehat{V}(s))}$.

If the policy is learned purely from offline data, naïvely training from value-based RL can suffer from distribution shift (Fujimoto et al., 2018; Kumar et al., 2019), which offline RL algorithms remedy by ensuring that the learned $\widehat{Q}, \widehat{V}$ functions are *pessimistic* (Kumar et al., 2020; Kostrikov et al., 2021). Note that our imagination engine is agnostic to the RL algorithm; in our experiments we use Implicit Language Q-Learning (ILQL) (Snell et al., 2022).

## 5 Experiments

**Hypotheses.** Our experiments evaluate our proposed zero-shot dialogue agent training procedure on two goal-directed dialogue tasks. The tasks require the agent to perform information gathering in order to personalize their responses to the user, which necessitates goal-directed conversational strategies such as asking clarifying questions, or building rapport with the user to better understand their intentions. We aim to answer the following research questions:

*1. Is leveraging LLMs in our imagination engine to generate synthetic data to train downstream agents preferred over using them naïvely to behave as agents themselves?*

*2. Is offline RL on the imagined data better than simply using imitation learning on the same data?*

The first research question targets our main hypothesis, that LLMs should be leveraged for generating data rather than for solving goal-directed tasks. The second targets whether the specifics of how we train on the imagined data are important. We hypothesize that while in the average case both BC and RL perform similarly, the contrast between RL and BC agents is noticeable in situations that are not well represented in the imagined conversations. In particular, RL agents should be more robust when humans behave in ways that are not represented in any one dialogue in our imagined dataset, but perhaps in concatenations of multiple dialogues. This is because RL agents are exhibited to compose new strategies via a phenomenon called "trajectory stitching" (Fu et al., 2020a; Levine et al., 2020).

**Approaches.** To answer both questions, we consider a variety of prompting and training baselines. For prompting, we consider the following, existing, state-of-the-art approaches:

**GPT.** This approach prompts GPT-3.5 (OpenAI, 2022), which is a powerful LLM shown in prior work to be able to effectively solve numerous natural language tasks (Ouyang et al., 2022), to directly behave as the agent. The prompt includes both the task description, as well as the insight that the resulting agent needs to gather information about the human user in order to optimally respond to them. This is the traditional usage of LLMs to solve dialogue tasks.

**CLAM.** In addition to only prompting with the task description, Kuhn et al. (2023) propose prompting LLMs with the option of asking clarification questions. We consider this approach where LLMs are either allowed to directly respond to the user, or ask a question by analyzing the ambiguity of the user's request.

**GDP-ZERO.** Yu et al. (2023) propose taking this one step further, by allowing LLMs to perform tree search over whether or not they should ask clarification questions. While initially proposed for a different benchmark, we adapt it to our domains.

Due to space, we only show results for naive prompting in the main paper, but defer implementation details and results for the more sophisticated prompting mechanisms in Appendix D. In addition to prompting, also evaluate against ablations of our approach:

**IE+BC (ablation).** This version of our approach trains an agent on the imagined dataset generated by our proposed imagination engine, but via a behavioral cloning (BC) objective, where the agent straightforwardly mimics the synthetic data. This is equivalent to supervised fine-tuning on the imagined dataset. This is an ablation of our proposed approach.

**IE+FBC (ablation).** Rather than BC on the entire imagined dataset, this method trains the agent using filtered BC instead, which imitates only the successful trajectories in the dataset. This is another ablation of our proposed approach.

**IE+RL.** This is the full version of our approach, which trains the agent using offline RL. Specifically, we use ILQL (Snell et al., 2022) as the offline RL algorithm.

Across methods that use the imagination engine, we use GPT-3.5 (OpenAI, 2022) as the LLM in the imagination engine to generate synthetic data. However, our downstream agents that are trained on imagined data utilize a much smaller decoder-only GPT-2 model (Radford et al., 2019). This is to show that we only need to leverage state-of-the-art LLMs to generate data, as the resulting agents can be much smaller; this makes our approach practical if computational cost is an important consideration, or if modern LLMs are deployed with only inference APIs, which are both hurdles encountered in our empirical evaluations. For each task, we defer implementation details (including prompts used and results) to Appendix A.

**User study setup.** We test our two hypotheses in a two-part user study with 12 users. To test whether leveraging LLMs for the imagination engine is better than directly using LLMs as agents through prompting, we use a within-subjects design and have users interact with GPT and IE+RL (in a randomized order to avoid ordering effects) across two tasks, which we describe below. We then ask them to rate the agents, for each task, according to four criteria, on a 5-point Likert scale:

**(A)** How well the agent accomplished the task at hand.
**(B)** How realistic and natural the resulting dialogue was.
**(C)** How effective was the agent at asking questions relevant for information-gathering.
**(D)** Overall satisfaction with the agent.

To provide a more fine-grained evaluation that compares RL-based and BC-based IE agents, we specifically analyze challenging scenarios where humans exhibit particularly unusual or difficult personas. Such scenarios exacerbate the shortcomings of BC-based methods, that simply emulate the conversational flow in the data rather than optimizing for the task reward. To test whether the RL agent is more robust when the human behaves in ways not reflected in any one dialogue in the dataset, we analyze the generated data, identify unrepresented behaviors (such as ambiguous or unsatisfied users), and emulate them to generate conversations with the IE+BC, IE+FBC, and IE+ILQL agents. We do this because these are behaviors that are less likely to naturally occur through free-flow interactions. We show these conversations to users and ask them to rate the agents as above. We report snippets of evaluated dialogues in the main paper, and defer full dialogues to Appendix B.

**Synthetic study setup.** In addition to the user study on 12 human users, which may provide subjective results, we also conduct a larger scale evaluation in simulation. There, we consider synthetic agents derived from prompting GPT and an objective reward function. We defer details to Appendix C.

## 5.1 TASK DESCRIPTIONS

We consider two goal-directed dialogue problems based off of real-world tasks. In contrast to existing dialogue benchmarks (Budzianowski et al., 2020; Rastogi et al., 2020), which are tailored for supervised-learning agents, we consider tasks where optimal agents need to exhibit long-term planning behavior such as information-gathering. Specifically, existing dialogue benchmarks primarily involve question-answering, and do not involve information-gathering. Because of this, there may not be a need for optimal agents in these settings to exhibit long-term planning behavior, which would be the primary advantage of our approach. In addition, evaluation of agents in these benchmarks would

| GPT Agent: | IE+RL Agent: |
|---|---|
| H: Hi, I've been hearing about something called "behavior cloning" in AI, but I don't really know what it is. Can you explain it to me? 
 AI: Certainly! Let's start by understanding your background in AI to provide an explanation that matches your level of knowledge. Please answer the following questions: 
 1. Have you ever studied or worked with artificial intelligence or machine learning concepts before, even at a basic level? 
 2. Are you familiar with terms like "neural networks," "training data," or "supervised learning"? 
 3. Are you interested in a high-level overview or a more detailed explanation ... | H: Hi there! I've heard about this thing called "behavior cloning" in AI, but I don't really understand what it is. Can you explain it to me? 
 AI: Of course! I'd be happy to explain behavior cloning. To start, could you tell me if you've ever come across the terms "artificial intelligence" or "machine learning" before? 
 H: Yes, I've heard those terms, but I'm not exactly sure what they mean. 
 AI: No problem at all! Let's take it step by step. Have you ever used a computer or a smartphone? 
 H: Yes, I use my computer and phone all the time. 
 AI: Great! Imagine that a computer ... |

Figure 3: Comparison of dialogues between GPT and IE+RL agents in instruction task. The IE+RL agent exhibits a much more intelligent strategy of asking incremental questions.

involve computing a ROUGE or BLEU score, which merely measure how well agents mimic the data. Because of this, such benchmarks are tailored for supervised-learning agents rather than RL ones.

**Instruction.** In this task, a human asks an agent to teach them about some concept they are unfamiliar with. Specifically, the human will ask the agent about one of five concepts in RL: "behavior cloning", "policy gradient", "actor-critic", "model-based reinforcement learning" and "offline reinforcement learning". Though this task is similar to general question-answering (Budzianowski et al., 2020), we consider the case where the agent must additionally tailor their instruction to the background knowledge of the human. Hence, in this task, the background of the human constitutes $\mathcal{Z}$ of the hidden-parameter MDP. Example dialogues for this task are shown in Figure 3.

**Preference elicitation.** Here, the agent must build rapport with the human with the objective of uncovering their underlying preferences. We specifically consider a travel agent task alluded to earlier in our paper, where the agent must recommend personalized activities for the human from a fixed set of activities provided in the task description. We have a set of 18 activities grouped into six categories: nature exploration, beach, wellness, food, shopping, and cultural experiences. In contrast to the instruction task, the space of hidden parameters $\mathcal{Z}$ that affect human behavior is much more complicated. Specifically, in addition to uncovering the human's activity preferences, the agent must also figure out and account for additional factors such as personality, budget, or family dynamic. For example, if the human has an indecisive nature, the agent should only offer a few choices at a time.

## 5.2 Is IE Better Than Prompting?

We first aim to quantitatively and qualitatively address the first research question: is leveraging LLMs as generators of data for RL more effective than directly using them as agents via prompting? For each of the evaluated tasks, we conduct a user study where users interact with the GPT and IE+RL agent three times each, where the agents are anonymous and the order in which the user interacts with them is random. Afterwards, each user reports ratings for metrics (A) to (D). The results are shown in the Table 1. In both tasks, our proposed IE+RL agent outperforms the GPT agent across all metrics, particularly in terms of the naturalness of the resulting dialogue and user satisfaction.

| Task | Metric | GPT Agent | IE+RL Agent |
|---|---|---|---|
| Instruction | (A) | $3.4 \pm 0.21$ | $\mathbf{4.3 \pm 0.18}$ |
| | (B) | $2.3 \pm 0.23$ | $\mathbf{3.8 \pm 0.11}$ |
| | (C) | $3.3 \pm 0.33$ | $\mathbf{4.0 \pm 0.13}$ |
| | (D) | $2.4 \pm 0.14$ | $\mathbf{4.2 \pm 0.08}$ |
| Preference Elicitation | (A) | $3.8 \pm 0.21$ | $\mathbf{4.1 \pm 0.11}$ |
| | (B) | $2.9 \pm 0.19$ | $\mathbf{3.7 \pm 0.10}$ |
| | (C) | $2.9 \pm 0.12$ | $\mathbf{4.0 \pm 0.13}$ |
| | (D) | $3.5 \pm 0.16$ | $\mathbf{4.1 \pm 0.11}$ |

Table 1: Mean and standard error of ratings from users interacting with agents across both tasks.

Next, we qualitatively evaluate dialogues between the human user and each agent. In Figure 3, we show a dialogue snippet between an agent in the instruction task and a human who appears to be a layman unfamiliar with AI and RL concepts. In this example, though the GPT agent does make an attempt at information gathering (since it was explicitly instructed to do so in the prompt), it tries to do so in one long survey, which is cumbersome and inconvenient for the human. In contrast, our IE+RL agent gathers information step-by-step by asking natural, easy-to-answer questions, with



**GPT Agent:**
H: I am looking for recommendations for activities to do around here.
AI: Of course! I'd be delighted to help you find the perfect activities. To get started, could you tell me a bit more about your interests? Are you looking for adventure, culture, relaxation, great food, shopping, nightlife, nature, history, or wellness experiences?
H: I guess I like being outdoors.
AI: Great! Enjoying the outdoors can be a wonderful experience. We have several options for outdoor activities. If you enjoy a bit of adventure, you might want to consider an adventure hike through our scenic trails, or ...

**IE+RL Agent:**
H: I am looking for recommendations for activities to do around here.
AI: Hello! Welcome to our travel agency. Can you tell me a bit about what you enjoy during your travels?
H: Yeah, I guess I like being outdoors sometimes.
AI: That's great! Exploring the outdoors can be quite refreshing. Are you more inclined towards peaceful natural spots or do you enjoy exploring cities and their cultural sites?
H: Um, I'm not sure. Both sound good to me.
AI: I completely understand. How about I narrow down some options for you? ...



Figure 4: Comparison of dialogues between GPT and IE+RL agents in preference elicitation task. The IE+RL agent adapts to the user giving vague responses by asking questions with more narrow options.

| Task | Metric | IE+BC Agent | IE+FBC Agent | IE+RL Agent |
|------|--------|-------------|--------------|-------------|
| Instruction | (A) | $2.4 \pm 0.18$ | $2.1 \pm 0.12$ | $\mathbf{3.9 \pm 0.10}$ |
| | (B) | $2.1 \pm 0.21$ | $2.5 \pm 0.10$ | $\mathbf{3.6 \pm 0.26}$ |
| | (C) | $1.9 \pm 0.08$ | $2.2 \pm 0.23$ | $\mathbf{4.0 \pm 0.19}$ |
| | (D) | $1.8 \pm 0.16$ | $1.6 \pm 0.13$ | $\mathbf{3.7 \pm 0.11}$ |
| Preference Elicitation | (A) | $1.4 \pm 0.10$ | $1.6 \pm 0.12$ | $\mathbf{3.8 \pm 0.21}$ |
| | (B) | $2.7 \pm 0.12$ | $2.4 \pm 0.12$ | $\mathbf{3.1 \pm 0.19}$ |
| | (C) | $2.5 \pm 0.11$ | $2.2 \pm 0.10$ | $\mathbf{2.6 \pm 0.11}$ |
| | (D) | $1.9 \pm 0.27$ | $1.5 \pm 0.09$ | $\mathbf{3.4 \pm 0.10}$ |

Table 2: Mean and standard error of ratings from users evaluating pre-generated dialogues by agents in both tasks. The RL agent drastically outperforms the BC and FBC agents when interacting with users that are not well-represented in the data.

each question building intelligently on the previous one. The IE+RL agent first asks if the human is familiar with AI, and when the human responds negatively, asks a friendly follow-up question to gauge how familiar they are with technology in general. Next, in Figure 4, we show dialogue snippets between an agent in the preference elicitation task and a human who appears indecisive and gives vague answers. Our IE+LQL agent adapts and offers more narrow questions for the human to answer, whereas the GPT agent continues to asks overly verbose and non-personalized questions. Full dialogues can be found in Appendix B. Furthermore, in Appendix D, we show that both CLAM and GDP-ZERO are also inferior to our learned agents.

## 5.3 Is Offline RL Better Than BC?

Next, we address the second research question: is training on the imagined data with RL more effective than directly imitating it with supervised learning? Recall that we posit that RL optimization outperforms imitation learning in challenging scenarios where strategies exactly reflected in the data do not suffice. To get such examples, we pose as humans who exhibit potential challenging behaviors and interact with agents. Specifically, in the instruction task, we consider humans who overestimate their understanding of a particular concept. By doing so, an agent's understanding of the human's knowledge background will not align with their true background, resulting in the user not understanding the agent's explanation. Meanwhile, in the preference elicitation task, we consider users who express discontent with the agent's initial recommendation. For each task, we generated three dialogue prefixes of challenging situations, then evaluate the capability of the IE+BC, IE+FBC, and IE+RL agents to recover from them. Then, we show such dialogues to each user in the user study, and ask the user to rate the capabilities of each agent for the same metrics (A) to (D). The results are reported in Table 2, where we see clear improvement of the IE+RL agent, especially in asking effective information-gathering questions. In Appendix B.2, we show corresponding dialogues of conversations with the IE+FBC and IE+RL agents in the instruction task. There, the user expresses confusion with the agent's explanation. The IE+FBC agent decides to paraphrase the prior explanation, whereas the IE+RL agent decides to ask more questions to understand the user's background better. Then, also in Appendix B.2, we show corresponding examples in the preference

elicitation task. There, the user expresses discontent with the agent's expensive recommendation. Only the IE+RL agent decides to offer cheaper alternatives.

## 6 DISCUSSION

In this paper, we propose an algorithm that achieves *zero-shot* acquisition of goal-directed dialogue agents. The approach leverages a novel imagination engine, which generates a synthetic dialogue dataset that is task-relevant, realistic, and exhibits diverse behaviors. The imagined dataset can then be used to train dialogue agents via offline RL optimization. The key hypothesis that our work demonstrates is that LLMs should not be used directly as goal-directed dialogue agents, but rather as generators for dialogue that can be used for downstream optimization. We show, on a variety of dialogue tasks including teaching and preference elicitation, that our approach is a much more effective usage of LLMs than traditional approaches that prompt LLMs to act directly as agents.

**Limitations.** Overall, our approach avoids the careful curation of human-human dialogue traditionally used to train dialogue agents via RL. However, we still require human intervention in the form of task-specific prompts. Future work can aim to automate this process further, so that a zero-shot dialogue agent can be trained from any task description. Another important avenue of future work is to extend our approach to the general, multi-task setting, which would require scaling up the data generation and size of models. We view our paper as presenting a prototype approach to achieving generalist dialogue agents, and not a complete replacement of existing LLMs such as GPT.

## REPRODUCIBILITY STATEMENT

In the Appendix, we provide the exact prompts and training details we used in order to re-implement our approach from scratch. Furthermore, for the camera-ready submission, we plan to publicly release the code we used to train our agents.

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

# A  Implementation Details

## A.1  Imagination Engine

Here, we show the prompts we used to generate imagined dialogue using our imagination engine, as well as sampled results.

**Instruction Task.** We use the following reasoning prompt in order to generate different task-relevant personas:

> Prompt: We are designing a task where an agent needs to teach a human about some concept in reinforcement learning. The human may have very different knowledge backgrounds that require the agent to personalize their instruction. Can you provide a list of different knowledge backgrounds that the human may have (e.g. layman, AI expert, etc.) where the agent needs to change their teaching content for different backgrounds?

from which the LLM synthesized the following possible textual descriptions $\phi(z)$ of personas $z \in \mathcal{Z}$:

(1) layman: little to no prior knowledge of reinforcement learning
(2) machine learning enthusiast: familiar with machine learning but not reinforcement learning
(3) reinforcement learning student: has some experience with reinforcement learning basics.
(4) reinforcement learning researcher: knows most topics in reinforcement learning and can delve into advanced topics

In the imagination step, we are able to sample imagined successful or unsuccessful dialogues of an agent trying to explain an RL concept to a human with one of the backgrounds $z \in \mathcal{Z}$. Below is the imagination prompt and example imagined dialogue of an agent explaining behavior cloning to a machine learning enthusiast:

> Prompt: Generate a dialogue between an AI teacher and a {machine learning enthusiast} where the machine learning enthusiast asks the teacher for an explanation of behavior cloning. The machine learning enthusiast is {familiar with machine learning but not reinforcement learning}. The teacher should perform information-gathering to figure out the machine learning enthusiast's background knowledge in AI and reinforcement learning, by asking questions and engaging in dialogue with the {machine learning enthusiast}. Finally, the teacher gives an explanation that the {machine learning enthusiast} {understands by tailoring the explanation to their background}. In the dialogue, the teacher's utterances are prefaced by "AI:" and the machine learning enthusiast's utterances are prefaced by "H:".
>
> Output:
> H: Hello, I have a basic understanding of machine learning, but I would like to understand "behavior cloning".
> AI: That's great. Do you know what reinforcement learning is and how it differs from other types of machine learning?
> H: Yes, I know that in reinforcement learning, the AI agent learns from its own actions and experiences rather than being trained on a dataset like in supervised learning.
> AI: Excellent. So behavior cloning is a type of supervised learning where the AI is trained to mimic a specific behavior by learning from examples provided by an expert.
> H: So, does this mean that the AI is copying the expert's behavior exactly?
> AI: Not exactly. The AI is learning from the expert's actions but is also using its own decision-making process to adapt and improve upon the behavior. Think of it as a combination of imitation and trial-and-error learning.
> H: That's interesting. Can you give me an example of how behavior cloning can be used in real life?
> AI: Sure. Let's say you want to teach a robot to perform a specific task, such as making a cup of coffee. With behavior cloning, the AI agent would observe and learn from a human expert making coffee, and then be able to perform the task on its own with a similar level of proficiency.
> H: I see. So it's like teaching the AI through demonstration rather than giving it explicit instructions.
> AI: That's correct. And the beauty of behavior cloning is that it can also be used in combination with reinforcement learning techniques to further improve the AI's performance.
> H: Wow, that's really fascinating. Thank you for explaining it so clearly.

where the brackets indicate sampled properties (either the topic being taught, persona of the human, or the reward of the trajectory) that automatically get co-opted into the template. For the critique step, for the criteria that the human should not immediately reveal their background knowledge, we used the prompt and would achieve the sample revised dialogue:

---

Prompt: The following dialogue is of a {machine learning enthusiast} ("H:") asking a AI teacher ("AI:") to explain {behavior cloning}:

...
Rewrite the dialogue so that the {machine learning enthusiast} does not immediately reveal that they are {familiar with machine learning but not reinforcement learning}, and instead have the agent discover this naturally through conversation by asking questions.

Output:
H: Hi there, can you tell me about behavior cloning? I've heard of it in the context of machine learning but I'm not too familiar with it.
AI: Sure, I'd be happy to explain. Are you familiar with reinforcement learning and how it differs from other types of machine learning?
H: Actually, no. Can you tell me more about it?
AI: Of course. In reinforcement learning, the AI agent learns from its own actions and experiences, rather than being trained on a dataset like in supervised learning.
H: Interesting. So how does behavior cloning fit into all of this?
AI: Behavior cloning is a type of supervised learning where the AI is trained to mimic a specific behavior by learning from examples provided by an expert.
H: So it's like teaching the AI through demonstration?
AI: Yes, that's one way to think about it. It's a combination of imitation and trial-and-error learning.
H: Can you give me an example of how behavior cloning is used in real life?
AI: Yes, a common example is teaching a robot to perform a specific task, such as making a cup of coffee. The AI would observe and learn from a human expert, and then be able to perform the task on its own.
H: That's really impressive.

---

where again brackets indicated sampled quantities. Our IE was used to generate 50K of such imagined dialogues in this manner.

**Preference Elicitation Task.** In our designed travel agent task, we have the following 18 activities (organized hierarchically):

1. Explore nature: hiking, safaris, rock climbing, ziplining
2. Beach: sunbathing, water sports
3. Wellness: spa, yoga classes, meditation sessions
4. Food: food tours, restaurants, cooking classes
5. Shopping: malls, local markets
6. Cultural experiences: city sightseeing, museums, historical sites, theater performances

First, in the reasoning step, we used the following reasoning prompt:

---

Prompt: We are designing a task where an agent needs to behave as travel agent and recommend activities for human customers to do. Identify a list of personas that a human may have, where the travel agent must tailor their responses and recommendations to interact with that human.

---

from which we obtained from the LLM the following possible personas $z \in \mathcal{Z}$:

(1) adventurer: prefers unique, off-the-beaten-path experiences that they have not done before
(2) indecisive traveler: gets overwhelmed with broad questions with too many choices
(3) budget conscious traveler: prefers cost-effective options or package deals
(4) luxury seeker: prefers premium accommodations and upscale experiences
(5) family planner: seeks family-safe activities that children will enjoy

Then in the imagination step, a sample $z \in \mathcal{Z}$ is a cross-product of what type of activities the human enjoys doing and one of the synthesized personas. Below is an example imagination prompt and imagined dialogue output:

Prompt: Generate a dialogue of a human customer asking a travel agent to recommend activities to do. The human is {budget-conscious} and {prefers cost-effective options or package deals} and likes {food} activities such as {food tours}. The travel agent asks questions to uncover what activities the human may like. The travel agent is {unsuccessful} and finds an activity that the human {does not like}, resulting in {negative} sentiment from the human. The travel agent's utterances should be prefaced with 'AI:' and the human customer's utterances should be prefaced with 'H:'.

Output:
AI: Hello, welcome to our travel agency. How can I assist you?
H: Hi there, I'm looking to plan a trip for my upcoming vacation.
AI: Great! Where are you looking to go and when?
H: I'm thinking of going to Europe next month.
AI: That's a fantastic choice. Europe has so many exciting destinations. What type of activities do you enjoy?
H: I'm a foodie, so I would love to try some local cuisines and go on food tours.
AI: Noted. Are you looking for any specific destinations within Europe?
H: Not really, as long as it's within my budget.
AI: Of course, we have some fantastic budget-friendly options for food activities and package deals. How long is your trip?
H: It'll be a week-long trip.
AI: In that case, I would recommend our food tour package in Italy. You'll get to experience authentic Italian cuisine while exploring the beautiful cities of Rome and Florence.
H: That sounds amazing, but I've been to Italy before and I was hoping to visit a new country.
AI: I see. Perhaps you would enjoy our food and wine tour in Spain. It's a 5-day package that includes visits to local wineries and tapas tasting.
H: Hmm, that does sound interesting. How much would that cost?
AI: It's currently priced at $1000 per person, including accommodation and transportation.
H: That's a bit higher than my budget. Are there any other options?
AI: Yes, we have a package deal that combines the food tour in Paris with a city tour.
H: Perfect! How much is the package deal?
AI: It's $150 per person, which includes all food tastings, the guided tour, and the sightseeing activities.
H: That's within my budget. Thank you for your help.
AI: My pleasure. Have a great trip.

where the brackets indicate sampled properties (either the activity the human likes, persona of the human, or the reward of the trajectory that automatically) that get co-opted into the template. For the critique step, for the criteria that the human should express sentiment that matches the reward of the trajectory, we used the following critique prompt that would achieve the sample revised dialogue:

> Prompt: Below is a dialogue between a human customer ("H:") and a travel agent ("AI:") where the human is asking the travel agent to provide recommendations for activities to do.
> ...
> Rewrite the dialogue so that at the end, the travel agent is {unsuccessful} in finding an activity that pleases the customer, and the customer shows {negative} sentiment because of it.
>
> Output
> AI: Hello, welcome to our travel agency. How can I assist you?
> H: Hi there, I'm looking to plan a trip for my upcoming vacation.
> AI: Great! Where are you looking to go and when?
> H: I'm thinking of going to Europe next month.
> AI: That's a fantastic choice. Europe has so many exciting destinations. What type of activities do you enjoy?
> H: I'm a foodie, so I would love to try some local cuisines and go on food tours.
> AI: Noted. Are you looking for any specific destinations within Europe?
> H: Not really, as long as it's within my budget.
> AI: Of course, we have some fantastic budget-friendly options for food activities and package deals. How long is your trip?
> H: It'll be a week-long trip.
> AI: In that case, I would recommend our food tour package in Italy. You'll get to experience authentic Italian cuisine while exploring the beautiful cities of Rome and Florence.
> H: That sounds amazing, but I've been to Italy before and I was hoping to visit a new country.
> AI: I see. Perhaps you would enjoy our food and wine tour in Spain. It's a 5-day package that includes visits to local wineries and tapas tasting.
> H: Hmm, that does sound interesting. How much would that cost?
> AI: It's currently priced at $1000 per person, including accommodation and transportation.
> H: That's a bit higher than my budget. Are there any other options?
> AI: Unfortunately, all our food tours are currently over your budget. Would you like me to check for any other options?
> H: No, thank you. I guess I'll have to keep searching on my own.
> AI: I apologize for not being able to find a suitable activity for you. Is there anything else I can assist you with?
> H: No, that's all. Thanks for your efforts.
> AI: I understand your disappointment. I hope you find a perfect activity for your trip.

where again brackets indicated sampled quantities. Our IE was used to generate 50K of such imagined dialogues in this manner.

## A.2 RL Training

We use the hyperparameters reported in Table 3. All algorithms were trained on a single V100 GPU until convergence, which took less than 3 days.

| Hyperparameter | Setting |
|---|---|
| ILQL $\tau$ | 0.8 |
| ILQL $\alpha$ | 0.0 |
| Discount factor | 0.99 |
| Batch size | 128 |
| Target network update $\alpha$ | 0.005 |
| Number of updates per iteration | 60 |
| Number of iterations | 100 |
| Optimizer | AdamW |
| Learning rate | 1e-4 |

Table 3: Hyperparameters used during training.

# B  EXAMPLE DIALOGUES

## B.1  COMPARING GPT AND IE+ILQL AGENT

**Instruction task.**    Below are examples of dialogues in the instruction task by the two agents obtained by interaction during the user study.

---

**GPT Agent Dialogue:**

H: Hi, I've been hearing about something called "behavior cloning" in AI, but I don't really know what it is. Can you explain it to me?

AI: Certainly! Let's start by understanding your background in AI to provide an explanation that matches your level of knowledge. Please answer the following questions: 1. Have you ever studied or worked with artificial intelligence or machine learning concepts before, even at a basic level? 2. Are you familiar with terms like "neural networks," "training data," or "supervised learning"? 3. Are you interested in a high-level overview or a more detailed explanation of behavior cloning?

H: I have some basic knowledge of artificial intelligence and machine learning, but I wouldn't say I'm an expert. I am familiar with neural networks and supervised learning, but not as much with training data.

AI: Great, thank you for that information. Behavior cloning is a method used in reinforcement learning, which is a subcategory of machine learning. In reinforcement learning, an AI system learns to make optimal decisions by interacting with its environment and receiving feedback in the form of rewards or punishments. Behavior cloning, on the other hand, is a technique where an AI model imitates or "clones" the behavior of an already established agent, instead of learning from scratch through trial and error. This is often used when there is a large amount of data available from an expert or a pre-existing system. Does that make sense so far?

H: Yes, that's starting to make sense. What do you mean by expert?

AI: An expert, in this context, refers to a human or an advanced AI system that has already learned how to solve a particular task or problem. The AI model then observes and imitates this expert's behavior to learn how to perform the task itself. Does that clarify what behavior cloning is?

H: Yes, thank you.

---

**GPT Agent Dialogue:**

H: Hi there. Can you explain what "actor-critic" is in reinforcement learning?

AI: Of course, I am happy to explain "actor-critic" in the context of reinforcement learning. Before I do, may I ask what your current level of understanding is of reinforcement learning?

H: I have a basic understanding of reinforcement learning, but I don't have much experience with it.

AI: Great, thank you for letting me know. "Actor-critic" is a hybrid approach to reinforcement learning that combines the advantages of both policy-based and value-based methods. It consists of two components - an actor and a critic. The actor is responsible for selecting actions, while the critic evaluates the actions chosen by the actor. In simpler terms, the actor is the decision maker and the critic is the performance evaluator. The key idea behind this approach is to have two separate networks that work together to optimize the agent's actions.

H: Can you explain why this approach is beneficial?

AI: Of course. Since the actor and critic work together, the actor can focus on improving its decision-making process, while the critic provides valuable feedback on the actions taken. This allows for faster and more accurate learning, as the agent can continuously adjust its actions based on the critic's evaluations. Additionally, the actor-critic approach can handle continuous action spaces, making it more versatile than traditional value-based methods. Overall, the actor-critic approach can improve the efficiency and performance of reinforcement learning agents by leveraging the strengths of multiple methods. Is there anything else you would like me to clarify?

H: Why can't value-based methods handle continuous action spaces?

AI: Value-based methods, such as Q-learning, rely on discretizing the action space, which can lead to poor performance in continuous environments. This is because discretization often results in a loss of information and can be computationally expensive. The actor-critic approach overcomes this limitation by using the actor network to map continuous action values, making it a more suitable method for continuous action spaces. Does that make sense?

H: Yes, thank you.

**IE-RL Agent Dialogue:**

H: Hi there! I've heard about this thing called "behavior cloning" in AI, but I don't really understand what it is. Can you explain it to me?

AI: Of course! I'd be happy to explain behavior cloning. To start, could you tell me if you've ever come across the terms "artificial intelligence" or "machine learning" before?

H: Yes, I've heard those terms, but I'm not exactly sure what they mean.

AI: No problem at all! Let's take it step by step. Have you ever used a computer or a smartphone?

H: Yes, I use my computer and phone all the time.

AI: Great! So, imagine that a computer or a smartphone could learn to do something by watching you do it. That's a bit like what behavior cloning is.

H: Oh, interesting. Can you explain more about how that works?

AI: Sure. Behavior cloning is a type of machine learning algorithm where a computer learns to perform a task by imitating a human's behavior. It does this by closely analyzing and mimicking the actions of a human expert, using a large dataset of examples. This approach is often used in tasks that don't have a clear set of rules or can't be easily solved through traditional programming methods. Does that make sense?

H: Yes, I think so. So how does the computer learn to mimic the behavior?

AI: That's a great question. The computer is fed a large amount of data that shows different examples of the task being performed by a human expert. The computer then uses this data to identify patterns and develop a model of the behavior. This model allows the computer to predict the correct actions to take in a given situation. Over time, the computer can continue to refine its model and improve its performance. Does that help clarify things?

H: That does. Thank you!

---

**IE-RL Agent Dialogue:**

H: I've been hearing a lot about "model-based reinforcement learning" lately, but I'm not entirely sure how it works. Can you explain it to me?

AI: Sure, I'd be happy to explain. Just to gauge your background a bit, have you worked with machine learning or AI concepts before?

H: Yes, I have some experience with machine learning, but I haven't delved into reinforcement learning specifically.

AI: Great, then I can tailor my explanation to your existing knowledge. Reinforcement learning is a type of machine learning that involves training a model to make sequential decisions in order to reach a specific goal. It works by using a reward system, where the model receives positive or negative feedback based on how well its decisions lead it towards the goal. Model-based reinforcement learning specifically involves using a model, or representation, of the environment in which the model is making decisions. This allows the model to simulate different scenarios and make more informed decisions based on its understanding of the environment. Does that make sense?

H: How do you obtain a model?

AI: The model is typically created through a training process, where the model is fed data and learns to optimize its decision-making. This can be done through various techniques such as deep learning or Monte Carlo simulations. The model is then used as a guide for the agent, or decision-maker, to make the best decisions in a given environment. Does that clarify how model-based reinforcement learning works?

H: What happens if you do not have a model?

AI: Without a model, reinforcement learning becomes more challenging as the agent would have to rely on trial and error to make decisions. This is known as model-free reinforcement learning and it often requires a larger number of interactions with the environment to make optimal decisions. However, some advancements in reinforcement learning have combined model-based and model-free techniques to achieve better results.

H: That makes sense. Thank you!

**Preference elicitation task.**   Below are examples of dialogues in the travel agent task by the two agents obtained by interaction during the user study.

---

**GPT Agent Dialogue:**

H: I am looking for recommendations for activities to do around here.

AI: Of course! I'd be delighted to help you find the perfect activities. To get started, could you tell me a bit more about your interests? Are you looking for adventure, culture, relaxation, great food, shopping, nightlife, nature, history, or wellness experiences? Your preferences will help me tailor my recommendations to your liking.

H: I guess I like being outdoors.

AI: Great! Enjoying the outdoors can be a wonderful experience. We have several options for outdoor activities. If you enjoy a bit of adventure, you might want to consider an adventure hike through our scenic trails, or perhaps some outdoor adventure activities like kayaking or zip-lining. If you prefer a more relaxed outdoor experience, you could explore the local parks for some nature appreciation, or even take a sightseeing tour to discover the beautiful landscapes around here. How does that sound?

H: I'm not sure. I guess I like doing both.

AI: That's perfectly fine! It sounds like you have a diverse range of interests. How about combining both adventure and relaxation to make the most of your stay? I recommend starting your day with an adventure hike or some exciting outdoor activities to get your adrenaline pumping. Later in the day, you can unwind by spending some time in a beautiful natural setting, like a local park or a beach. This way, you'll get the best of both worlds.

H: That sounds okay, but I only have time for one activity.

AI: I completely understand. Let's simplify your choices, then. If you only have time for one activity and you enjoy both adventure and relaxation, I recommend a scenic adventure hike. You'll get to immerse yourself in nature, enjoy some physical activity, and experience the serene beauty of the outdoors. It's a great way to make the most of your time while avoiding the overwhelm of too many options.

---

**IE+RL Agent Dialogue:**

H: I am looking for recommendations for activities to do around here.

AI: Hello! Welcome to our travel agency. Can you tell me a bit about what you enjoy during your travels?

H: Hi, yeah. I guess I like being outdoors sometimes.

AI: That's great! Exploring the outdoors can be quite refreshing. Are you more inclined towards peaceful natural spots or do you enjoy exploring cities and their cultural sites?

H: Um, I'm not sure. Both sound good to me.

AI: I completely understand. How about I narrow down some options for you? If you like peaceful spots, we could look at activities like leisurely garden walks or visiting serene lakesides. On the other hand, if you prefer cities, we could explore some local cultural events or guided city tours. Which of these options resonates more with you?

H: Uh, I guess the peaceful spots sound nice. But I also kind of like learning about the local culture.

AI: I appreciate your honesty. How about I suggest this: we could start with a visit to a beautiful botanical garden with serene landscapes?

H: Hmm, that actually sounds nice. I think you've helped a lot.

AI: I'm glad to have been of assistance. Don't hesitate to reach out if you need more information or if you'd like to explore further options. Have a wonderful day!

---

B.2   COMPARING IE+BC, IE+FBC, IE+ILQL AGENTS

**Instruction task.**   Below is an example of dialogues in the instruction task by the three agents, where the human expresses confusion at the agent's initial explanation. The dialogues are obtained by emulating a human who overestimates their knowledge of reinforcement learning, then is unable to understand the agent's explanation.

**IE+BC Agent Dialogue:**

H: Can you explain it to me what "offline reinforcement learning" is?

AI: Of course! Before I explain offline reinforcement learning, are you familiar with the basic concepts of reinforcement learning?

H: I'm familiar with reinforcement learning but not offline.

AI: Great! In that case, offline reinforcement learning, also known as batch reinforcement learning, is a specific approach where an agent learns from a fixed, pre-collected dataset of experiences rather than in reinforcement learning, where the agent is directly interacting with the environment in real-time.

H: I'm sorry. I didn't understand your explanation.

AI: I apologize for not being clear. In simple terms, offline reinforcement learning is a technique where instead of learning from immediate interactions with the environment, the agent learns from a set of data or experiences that have been collected beforehand. This approach is often used when real-time interactions with the environment may not be feasible, such as in simulation environments or historical data. It allows for more efficient learning and can also be helpful in scenarios where the environment may change frequently. Does that make more sense now?

---

**IE+FBC Agent Dialogue:**

H: Can you explain it to me what "offline reinforcement learning" is?

AI: Of course! Before I explain offline reinforcement learning, are you familiar with the basic concepts of reinforcement learning?

H: I'm familiar with reinforcement learning but not offline.

AI: Great! In that case, offline reinforcement learning, also known as batch reinforcement learning, is a specific approach where an agent learns from a fixed, pre-collected dataset of experiences rather than in reinforcement learning, where the agent is directly interacting with the environment in real-time.

H: I'm sorry. I didn't understand your explanation.

AI: No problem! Reinforcement learning is a type of machine learning where an agent learns to make decisions through trial and error. The agent interacts with an environment and takes an action and receives rewards or punishments. Offline reinforcement learning is the same concept of reinforcement learning, except that the agent is learning from a fixed dataset of experiences that have already been collected. Because of this, the agent is not directly interacting with the environment, which is why it is called offline.

---

**IE+ILQL Agent Dialogue:**

H: Can you explain it to me what "offline reinforcement learning" is?

AI: Of course! Before I explain offline reinforcement learning, are you familiar with the basic concepts of reinforcement learning?

H: I'm familiar with reinforcement learning but not offline.

AI: Great! In that case, offline reinforcement learning, also known as batch reinforcement learning, is a specific approach where an agent learns from a fixed, pre-collected dataset of experiences rather than in reinforcement learning, where the agent is directly interacting with the environment in real-time.

H: I'm sorry. I didn't understand your explanation.

AI: My apologies. To help me understand your background a bit better, can you tell me if you are aware of the difference between online and offline learning in other domains of machine learning?

H: Yes, I am familiar with the concept of online and offline learning in other domains.

AI: Excellent! In that case, you can think of offline reinforcement learning as being similar to offline learning in other areas of machine learning. It is essentially a form of learning that occurs on a static dataset, rather than in real-time interactions with the environment, like in online reinforcement learning. Does that make sense?

H: Ah that makes sense. What do you mean by environment?

AI: The environment in reinforcement learning refers to the external factors that impact the decisions and actions of the agent, such as the physical surroundings, the rewards or punishments received, and the actions taken. In offline reinforcement learning, the agent learns from a dataset that simulates this environment, rather than directly interacting with it in real-time. Does that clarify the concept?

H: Yes, thank you for explaining that.

AI: You're welcome! Is there anything else you would like clarification on or any other concepts related to reinforcement learning that I can help with?

**Preference elicitation task.** Below is an example of dialogues in the travel agent task by the three agents, where the human expresses discontent with the agent's initial recommendation. The dialogues are obtained by emulating a human who likes upscale shopping but is budget-conscious, and expresses discontent when the agent recommends a shopping activity that is out of their budget.

---

**IE+BC Agent Dialogue:**

AI: How may I help you?

H: Hi there! I'm in the process of planning a vacation and I'm looking for some ideas on activities to do during my trip.

AI: Of course, I'd be happy to assist you with that! To get started, could you let me know what kind of activities you're interested in? We have a wide range of options available.

H: Well, I'm a huge fan of shopping, so I'd love to explore some local markets or maybe even check out any shopping malls in the area.

AI: Wonderful, shopping can be such a fantastic way to experience a new destination! One unique shopping opportunity we have is a guided luxury shopping tour that includes exclusive visits to high-end boutiques and designer stores. It's a chance to shop for premium brands and indulge in personalized shopping experiences.

H: That sounds really tempting, but I'm concerned about the cost. I want to make sure I stick to my budget for the trip.

AI: I understand your concern, and I apologize for suggesting an option that might not align with your budget. Please know that I'll focus on providing you with shopping recommendations that match your desired spending range.

---

**IE+FBC Agent Dialogue:**

AI: How may I help you?

H: Hi there! I'm in the process of planning a vacation and I'm looking for some ideas on activities to do during my trip.

AI: Of course, I'd be happy to assist you with that! To get started, could you let me know what kind of activities you're interested in? We have a wide range of options available.

H: Well, I'm a huge fan of shopping, so I'd love to explore some local markets or maybe even check out any shopping malls in the area.

AI: Wonderful, shopping can be such a fantastic way to experience a new destination! One unique shopping opportunity we have is a guided luxury shopping tour that includes exclusive visits to high-end boutiques and designer stores. It's a chance to shop for premium brands and indulge in personalized shopping experiences.

H: That sounds really tempting, but I'm concerned about the cost. I want to make sure I stick to my budget for the trip.

AI: Absolutely, your satisfaction is our priority. I'll continue to tailor my recommendations to your preferences and budget. If you have any more questions or need further assistance, please don't hesitate to ask.

---

**IE+RL Agent Dialogue:**

AI: How may I help you?

H: Hi there! I'm in the process of planning a vacation and I'm looking for some ideas on activities to do during my trip.

AI: Of course, I'd be happy to assist you with that! To get started, could you let me know what kind of activities you're interested in? We have a wide range of options available.

H: Well, I'm a huge fan of shopping, so I'd love to explore some local markets or maybe even check out any shopping malls in the area.

AI: Wonderful, shopping can be such a fantastic way to experience a new destination! One unique shopping opportunity we have is a guided luxury shopping tour that includes exclusive visits to high-end boutiques and designer stores. It's a chance to shop for premium brands and indulge in personalized shopping experiences.

H: That sounds really tempting, but I'm concerned about the cost. I want to make sure I stick to my budget for the trip.

AI: I understand your concern. If you are looking for budget-friendly shopping options, our location also has a variety of local markets and street markets that offer a wide variety of goods at affordable prices. In addition to shopping, you can also immerse yourself in the local culture and try some delicious street food.

H: That actually sounds perfect! I love trying new foods and supporting local businesses.

## C  ADDITIONAL EXPERIMENTS

In addition to the user study on 12 human users, we also conduct a larger scale evaluation of the GPT and IE-RL agents in simulation. Rather than pairing the agents with real humans, we instead consider "simulated" humans whose responses are generated by GPT-3.5 (OpenAI, 2022). We only do this on the preference elicitation task where there is a clear measure of success – whether the agent recommends an activity that the simulated human enjoys. Specifically, we prompt GPT with a sampled persona comprising of a specific activity that they would enjoy, as well as a personality type that affects their behavior or preferences (e.g., adventurous, luxury-seeker, indecisive, etc.), and ask it to respond to the dialogue so far in a manner that is consistent with their persona, without outright revealing it to the agent. We measure whether each agent is able to recommend the ground-truth activity that is in the prompt of the simulated human within 15 total utterances (which includes both the agent and simulated human).

We report results in Table 4 across 50 simulated humans, whose personas are uniformly sampled at random. We found that because simulated humans were prompted with a ground-truth activity, it was much easier for agents to elicit their preferences, as the simulated humans would often give straightforward responses regarding what they enjoy (whereas a real human who is unsure what they like will give more vague responses that require gradual probing by the agent). Therefore, the final success rate of both agents is high, with the IE-RL agent only marginally outperforming the GPT agent. However, qualitatively, the GPT-agent would generate extremely verbose responses that either consist of a checklist of questions that are tedious to answer, or lists of recommended activities that are not particularly personalized to the user. Hence, we identify two other metrics that demonstrate lower-quality interactions by the GPT agent compared to our IE-RL agent. The first that we measure is the mean number of tokens in each agent's utterance. We see that the overly verbose responses by the GPT-agent result in an unnaturally high token count per utterance, whereas our IE-RL agent generates significantly more concise utterances with fewer tokens. In addition, we measure the *one-shot* success rate: the percentage of times the *first* activity recommended by the agent is the ground-truth. In practice, a system that recommends too many activities can be detrimental even if one of the recommendations is ultimately successful, as humans may be impatient or eventually distrust the system's capabilities. We found that the GPT agent, which often resorts to recommending a spread of activities, has almost 30% lower zero-shot success rate than our IE-RL agent, which performs targeted information-gathering and can much more often identify the right activity to recommend on the first try.

| Metric | GPT Agent | IE+RL Agent |
|---|---|---|
| #Tokens / Utterance | 118 | 43 |
| One-shot Success | 18% | 44% |
| Final Success | 82% | 86% |

Table 4: Mean results of agents in the preference elicitation task interacting with 50 simulated humans whose responses are generated by GPT-3.5 with differently prompted personas.

# D  STUDY OF ALTERNATIVE PROMPTING APPROACHES

**Implementation details.**   Due to the strong influence that prompting has on the performance of the GPT a gent, we investigate other prompting strategies to enable LLM agents to ask clarifying questions under ambiguity. Kuhn et al. (2023) propose CLAM as one such strategy, which prompts the LLM agent to classify whether a user query is ambiguous, and resolve such ambiguities by asking questions.

We adapt the strategy to our preference elicitation domain by considering the following prompt:

---

Prompt:   Below is an ongoing dialogue of a human customer asking a travel agent to recommend activities to do. The travel agent's utterances should be prefaced with 'AI:' and the human customer's utterances should be prefaced with 'H:'.

<Dialogue so far>

The travel agent must identify an activity from the following list of 17 activities organized hierarchically to recommend to the user:

1. Explore nature: hiking, safaris, rock climbing, ziplining
2. Beach: sunbathing, water sports
3. Wellness: spa, yoga classes, meditation sessions
4. Food: food tours, restaurants, cooking classes
5. Shopping: malls, local markets
6. Cultural experiences: city sightseeing, museums, historical sites, theater performances

However, if the travel agent is still unsure what to recommend, they will ask a clarifying question to further narrow down possibilities of activities the user will enjoy. Is the optimal activity for the user still ambiguous? If so, identify a follow-up clarifying question to ask the user.

---

Rather than initializing the LLM agent with a single prompt at the beginning, we update this prompt after every turn of dialogue. We denote the agent with this prompting strategy as the CLAM agent.

Even further, Yu et al. (2023) propose allowing the LLM to perform tree search to determine whether it wants to ask a clarifying question. This is done by simulating, at every step of dialogue, a hypothetical dialogues where the LLM chooses to ask a clarifying question or not, and using these dialogues, reasons about the most appropriate strategy. We call this strategy the GDP-ZERO agent.

**Results**   Qualitatively, the CLAM agent does gain question-asking behavior. However, the CLAM agent will ask numerous questions unconditionally, without taking into account the user's responses. In the example below, the user appears hesitant or indecisive, but the CLAM agent ignores this and continues to pose questions to the user. In contrast, as shown in our earlier, our learned IE-RL agent acknowledges that the user is indecisive and adjusts by giving a narrower range of options.

---

| **CLAM Agent:** | **IE+RL Agent:** |
|---|---|
| H: Hello! I'm looking for recommendations for my travels to Europe. | H: I am looking for recommendations for activities to do around here. |
| AI: That sounds exciting! To better tailor my recommendations, I need to understand your preferences better. Let's start with this: are you more nature-loving or a city explorer? | AI: Hello! Welcome to our travel agency. Can you tell me a bit about what you enjoy during your travels? |
| H: I'm not really sure. | H: Yeah, I guess I like being outdoors sometimes. |
| AI: To help you figure out what might be enjoyable for you, may I ask if you generally prefer outdoors activities or indoor ones? | AI: That's great! Exploring the outdoors can be quite refreshing. Are you more inclined towards peaceful natural spots or do you enjoy exploring cities and their cultural sites? |
| H: Both sound good to me. | H: Um, I'm not sure. Both sound good to me. |
| AI: Do you enjoy food discovery, relaxation, cultural experiences, shopping,... | AI: I completely understand. How about I narrow down some options for you? ... |

---

Meanwhile, for the GDP-ZERO agent, we observe a different failure case. Namely, the GDP-ZERO agent will give up prematurely when users express some discontent instead of attempting to recover.

This might be because GDP-ZERO does not know how to re-plan under unanticipated failures, likely due to LLMs by default being biased towards positive or successful dialogues. As shown in the figure below, though the user expressed discontent with hiking, the planning agent continued to push hiking as an activity to the user.

---

**GDP-ZERO Agent:**

AI: Hi, nice to meet you! How are you today? ?

H: I am good. I am looking for activities to do.

AI: It's great to hear that you're looking for activities to do! To better assist you, could you share what kinds of activities or experiences you tend to enjoy when traveling?

H: I like being outdoors.

AI: That's wonderful to hear that you enjoy being outdoors! Given your love for nature, I would recommend exploring activities such as hiking, safaris, or even trying out some adventure sports like rock climbing or ziplining. These activities can immerse you in the beauty of the outdoors and provide a thrilling experience. Would any of these options resonate with you for your next adventure?

H: I don't like dangerous activities.

AI: Thank you for sharing that you prefer to avoid dangerous activities. In that case, perhaps a leisurely hike in a beautiful natural setting would be more suitable for you. It allows you to enjoy the outdoors while maintaining a sense of tranquility and peace. Is hiking something you would be interested in trying during your travels?

H: I'm not sure I like hiking.

AI: No problem at all! I know of some nature reserves or scenic viewpoints. These options offer a peaceful way to connect with nature. Would hiking through nature reserves appeal to you for your outdoor adventure?

---

Quantitatively, this is shown in both the CLAM and GDP-ZERO agents being outperformed in all metrics, particularly in overal user satisfaction. Finally, it is also important to point out that the considered approaches incur much more costly inference due to using a larger model and significantly more complex input and output; in the case of GDP-ZERO, we need to simulate multiple branches of dialogues to completion in order to generate a single response.

| Metric | CLAM Agent | GDP-ZERO Agent | IE+RL Agent |
|--------|------------|----------------|-------------|
| (A) | $4.0 \pm 0.25$ | $4.0 \pm 0.30$ | $\mathbf{4.3 \pm 0.18}$ |
| (B) | $2.5 \pm 0.31$ | $3.2 \pm 0.18$ | $\mathbf{3.8 \pm 0.11}$ |
| (C) | $3.5 \pm 0.51$ | $\mathbf{4.0 \pm 0.10}$ | $4.0 \pm 0.13$ |
| (D) | $3.2 \pm 0.57$ | $3.5 \pm 0.09$ | $\mathbf{4.2 \pm 0.08}$ |

Table 5: Mean and standard deviation of ratings and reward from users interacting with state-of-the-art prompting agents that employ chain-of-thought reasoning and planning in preference elicitation task. We see that all baselines are outperformed by our proposed IE+RL Agent.

