# OpenReview forum: "Zero-Shot Goal Dialogue via Reinforcement Learning on Imagined Conversations"
_ICLR.cc/2025/Conference — Submitted to ICLR 2025_

### Official Review · Reviewer_m1Jy · 2024-10-29

**Soundness:** 2
**Presentation:** 3
**Contribution:** 2
**Rating:** 3
**Confidence:** 4

**Summary:**

This paper explores a new method for adapting large language models (LLMs) for goal-directed dialogues using reinforcement learning (RL). The key innovation in this work is the introduction of an "imagination engine," which synthesizes hypothetical human-human interactions based on task descriptions. These imagined dialogues serve as training data for offline RL, enabling the creation of conversational agents that can optimize multi-step objectives and gather information effectively. The proposed approach shows improved performance in tasks such as teaching and preference elicitation compared to traditional methods that use LLMs directly.

**Strengths:**

1. The introduction of an "imagination engine" to synthesize hypothetical dialogues is a novel approach. It creatively leverages LLMs' ability to generate diverse and human-like conversations.

2. This method adopts a multi-step optimization strategy to obtain better quality data.

**Weaknesses:**

1. Although the author mentioned efficiency considerations, it's somewhat difficult to justify using GPT-2 as the base model for experiments in this day and age. Why not try LLaMA or other more powerful open-source models?

2. The evaluation relies solely on human assessment, which is subjective. It would be better to incorporate objective evaluation metrics as a supplement. One possible approach could be to set aside around 10% of the dataset as a test set, run tests on it, and use metrics like BLEU and ROUGE to evaluate model performance. While this may not be the optimal solution, it’s better than nothing.

**Questions:**

1. I wonder why not introduce the criteria from the Critique Step during the Imagination Step? Wouldn't that make the process more streamlined?

2. I'm curious about the size of the synthesized dataset. Was it entirely used for RL training?

3. I would like to know the size of the test set used in the experiments. Additionally, I noticed that the evaluation was conducted by 12 different individuals. Is there any consistency check performed?

4. The authors assume that "models trained with RL outperform those using prompts" and conducted experiments with GPT-3.5. I am interested in knowing the exact prompt used to call the model, as it significantly affects the outcome of prompting. Moreover, the authors might consider conducting experiments with more advanced models (such as GPT-4o). Relying solely on GPT-3.5 does not strongly support the assumption, as its performance lags behind and may even fall short of some of the cutting-edge open-source models.

---

### Official Review · Reviewer_k2QM · 2024-11-03

**Soundness:** 2
**Presentation:** 2
**Contribution:** 1
**Rating:** 3
**Confidence:** 4

**Summary:**

The paper presents a new approach for training goal-directed dialogue agents by applying reinforcement learning (RL) to synthetic data generated from large language models (LLMs). While LLMs excel in general text generation, they often struggle with tasks requiring multi-turn, goal-oriented interactions. This study introduces an "Imagination Engine" (IE) that synthesizes realistic task-specific dialogues, which are then used to train RL-based agents capable of optimizing for outcomes in conversations. The approach is demonstrated on tasks like teaching concepts and eliciting user preferences, with experimental results indicating that the method outperforms direct prompting of LLMs in achieving conversational goals.

**Strengths:**

1. The method creatively leverages LLMs to generate diverse, goal-directed dialogues, addressing data scarcity in training agents for complex conversational tasks.
2. The paper shows an efficient application of offline RL by using synthetic dialogues, enabling scalable agent training without the need for real-time user interactions.
3. Empirical results, including user studies, suggest that the proposed method improves outcomes over conventional LLM-based approaches in teaching and preference elicitation tasks.

**Weaknesses:**

1. The authors use the term "goal-directed dialogue," but in NLP, the terms target-driven conversation and proactive dialogue are more widely used to describe similar tasks. These areas have established research and methods that could deepen the paper's connection to prior work.
2. The idea of using LLM to simulate conversations and then leverage offline reinforcement learning to train a model is not new. The authors might want to compare with a rather similar work here: https://aclanthology.org/2024.acl-long.262/
3. The evaluation is primarily in synthetic settings, limiting insights into how well the approach would perform in more dynamic, real-world user interactions with diverse needs.

**Questions:**

As detailed in weaknesses.

**Details Of Ethics Concerns:**

N.A.

---

### Official Review · Reviewer_YRaE · 2024-11-03

**Soundness:** 3
**Presentation:** 3
**Contribution:** 2
**Rating:** 5
**Confidence:** 4

**Summary:**

This paper proposes a novel method to train goal-directed dialogue agents using zero-shot RL. The core idea is to leverage LLMs to simulate human-like conversations, creating a diverse dataset, which is then used with offline RL to optimize dialogue agents for multi-step, goal-directed interactions. Experiments show that using LLMs to generate data and then training RL agents outperforms directly using LLMs as dialogue agents.

**Strengths:**

The imagination engine creates a varied dialogue dataset without requiring extensive human-collected data.

Combined with RL, the agents are trained to ask clarifying questions and make goal-directed decisions over multiple turns.

User studies show RL-based agents excel in natural dialogue flow and effective information gathering compared to traditional LLM methods​

**Weaknesses:**

The synthetic dataset generated by IE is based on LLM simulations, which may not fully reflect actual user behavior. Particularly for highly personalized or complex tasks, synthetic dialogues can diverge significantly from reality, as simulated users may appear overly cooperative or lack the randomness typical of real users. This discrepancy can affect the agent's performance in real-world scenarios.

Training with offline RL on a synthetic dataset can encounter the "distribution shift" problem, where the real-world dialogues that the agent encounters differ from the distribution of the training data. This mismatch may lead to poor performance when the agent faces novel scenarios. Although optimistic estimation techniques were applied to mitigate this, such methods cannot entirely eliminate the impact of distribution shifts.

Current evaluations are based on annotations from 12 users, which is a limited sample size and could introduce bias. Using the number of turns can indicate effectiveness, while satisfaction could be evaluated through various system assessment methods in dialogue systems. Larger, more reliable evaluation results would be beneficial.

While offline RL methods allow for policy optimization on fixed synthetic datasets, the absence of real-time feedback in dynamic and complex dialogue scenarios can lead to suboptimal strategies. For example, in real dialogues, user feedback or sentiment may change dynamically, and a fixed dataset cannot capture this variability fully, limiting the agent's adaptability and flexibility during actual interactions.

Since synthetic data is generated by large language models, it may lack real-world noise and complexity, particularly in ambiguous or conflicting user input. This lack of realistic data could lead to "over-idealized" behavior, meaning the agent may perform well in "clear and cooperative" scenarios but struggle when confronted with the unpredictability of actual users.

Some research on dialogue uncertainty also approaches the issue from an information-gathering perspective. The authors might consider comparing more advanced prompting methods with the current RL approach, as RL data collection and training costs are still relatively high.

-- Uncertainty of Thoughts: Uncertainty-Aware Planning Enhances Information Seeking in Large Language Models. https://arxiv.org/abs/2402.03271

-- MEDIQ: Question-Asking LLMs for Adaptive and Reliable Clinical Reasoning. https://arxiv.org/abs/2406.00922

**Questions:**

See the weaknesses

---

### Official Review · Reviewer_bjzC · 2024-11-06

**Soundness:** 2
**Presentation:** 3
**Contribution:** 2
**Rating:** 3
**Confidence:** 4

**Summary:**

This paper describes an approach for training  goal-directed dialog agents by leveraging synthetic data generated from LLM. The authors showed that agent trained on the LLM generated synthetic data has a higher performance than prompting LLM to act directly as an agent. TC also discussed the effectiveness of using behavior cloning vs. RL for training such agents.

**Strengths:**

1. This paper is well written and easy to follow. The author discussed two key hypotheses (the effectiveness of LLM trained on self generated synthetic data vs. direct prompting; and offline RL vs. behavior cloning), and used the same throughout the paper in the methodology and experiment sections which make it easy to comprehend and follow.
2. The proposed method is discussed in good detail. The authors presented the imagination engine and the RL optimization with good clarity. The authors provided provide comprehensive discussion on related work and preliminaries on MDP and RL which helped the presentation of the proposed method.
3. Comprehensive experiments against multiple baseline methods. The authors compared the proposed method to different baselines on multiple tasks to illustrate the effectiveness of the proposed method. The authors also provided detailed examples to showed the quality of the responses from different approaches.

**Weaknesses:**

1. The authors made some vague and strong claims in the paper that are not well supported. e.g. line 76 “In effect, the LLM can imagine what a human could do, but not to what an optimal agent should do”; line 250-253 “Since inferring the human’s persona is an important skill we want downstream learning agent to acquire”.
2. The quality of the synthetic data produced by the “imagination engine”, which plays a key role in the optimization of the dialog agent through RL, is not sufficiently discussed. For example, the author sampled reward score, and used that as part of the input for the synthetic dialog generation. It’s unclear how closely the LLM followed the instruction in generating the dialogs. Without understanding the quality of the generated data, it’s hard assess the effectiveness of the optimization with RL.
3. Training dialog agent using offline RL from dialog corpus is not something new. It has been widely explored in dialog research literatures. The main novelty of the work to me is on leveraging self-generated synthetic data for RL training. To strengthen the argument that this is an effective approaching comparing to prompting LLMs directly, I would expect the authors to discuss more on the intuition of this approach and the corresponding validation, in addition to the experiment results on response quality.

**Questions:**

1. What's the quality of the synthetic data?
2. What's the intuition that training the dialog agent on self generated data works better than prompt the llm directly?
3. Line 277: r = r_i only if s' = \tau_i is the full dialog - what's the assigned value of r when it is not the end of the dialog?

---

### Meta-Review · Area_Chair_BrXp · 2024-12-17

**Metareview:**

This work proposes a method to generate synthetic data for goal-oriented agents, leveraging LLMs to simulate human conversations. The synthetic data is then used with offline RL to train a goal-oriented conversational agent. The reviewers appreciate the clarity of the paper as well as the thorough evaluation and some empirical results. However, they also raise several concerns, such as unsupported claims, the overall novelty of the approach, as well as limited scale of human evaluations.

**Additional Comments On Reviewer Discussion:**

No discussions, the authors did not provide a response to the reviews.

---

### Decision · Program_Chairs · 2025-01-22

Reject